# Sweep Frequency Heating based on Injection Locked Magnetron

**Fengming Yang** [1,2], **Wenwen Wang** [1,2], **Bo Yan** [3,4], **Tao Hong** [5], **Yang Yang** [1,2], **Huacheng Zhu** [1,2,*], **Li Wu** [1,2] and **Kama Huang** [1,2]

1 College of Electronic and Information Engineering, Sichuan University, Chengdu 610065, China; 2017222055248@stu.scu.edu.cn (F.Y.); yujie-cd@163.cm (W.W.); yyang@scu.edu.cn (Y.Y.); wuliscu@126.com (L.W.); kmhuang126@126.com (K.H.)

2 Key Laboratory of the Wireless Power Transfer, Ministry of Education, Sichuan University, Chengdu 610065, China

3 College of Chemistry and Chemical Engineering, Guizhou University, Guiyang 550025, China; yanbo993@126.com

4 State Key Laboratory of Efficient Utilization for Low Grade Phosphate Rock and Its Associated Resources, Wengfu (Group) Co., Ltd., Guiyang 550014, China

5 School of Electronic Information Engineering, China West Normal University, Nanchong 637002, China; scu_mandela@163.com

* Correspondence: hczhu@scu.edu.cn; Tel.: +86-28-8547-0659

**Abstract:** Conventional microwave heating has serious problems such as non-uniform heating and low efficiency. A novel magnetron microwave sweep frequency heating method is proposed to improve microwave heating uniformity. In this method, the frequency-sweeping signal is injected into the magnetron by the injection frequency-locking technique, and the microwave sweep frequency heating of the magnetron is realized. In this paper, a complicated injection frequency locking system is given and analyzed and a multiphysics calculation model based on the finite element method for electromagnetic waves and heat transfer is established. The calculation of microwave sweep frequency heating is realized by the combination of COMSOL and MATLAB. The results show that the sweep frequency heating has an obvious superiority. An experiment is carried out to verify the simulation results. The simulation results are in agreement with the experimental data. Moreover, the effect of sweep bandwidth and sweep interval on heating uniformity is discussed.

**Keywords:** injection-locking; sweep frequency; heating uniformity; microwave heating

## 1. Introduction

In recent decades, microwave energy has been widely used as a highly efficient clean energy in various fields such as food processing, the chemical industry, and medicine [1–3]. Microwaves interact with polar molecules and charged ions. The friction resulting from molecule alignment and migration of charged ions in rapidly alternating electromagnetic field generates heat within matter. Microwave can penetrate the inside of the material to heat, and achieve the effect of simultaneous heating inside and outside [4]. Compared with the traditional heating method, microwave heating is characterized by efficient heating, internal heating, and environmentally friendly heating.

However, due to the uneven distribution of the electromagnetic field in the cavity, there are also a series of problems such as poor heating uniformity, more hot spots and thermal runaway, etc. [5]. These problems have greatly restricted the application of microwave energy in the industry. Therefore, great efforts have been made by experts to invent new methods to improve heating uniformity in the microwave industry.

In order to solve the problem of poor microwave heating uniformity, many scholars and experts have done a lot of research in this area. Increasing the uniformity of microwave heating is also improving the uniformity of the energy distribution of the electromagnetic field in the reaction cavity. The mode or distribution of the electric field may be changed drastically during the whole heating process, thereby improving the uniformity of the electromagnetic field energy during the whole heating process [6]. For microwave single-mode heating of the reaction cavity, the most common method of improving heating uniformity is to change the frequency and phase of the microwave. For the microwave multi-mode reaction cavity, the uniformity of heating can be improved by changing the mode of electromagnetic waves or using different frequencies. The most common method of changing the mode of electromagnetic waves is to use a mode stirrer [7–9]. When the microwave frequency is fixed, we can change the position and shape of the microwave feed port [10], add a metal stirrer to the cavity [11], or add a transmission device to the heated object to change the electromagnetic field mode [12]. When the microwave frequency changes, different frequencies correspond to different electromagnetic field modes [13]. The uniformity of heating may be also improved.

Václav et al. used numerical analysis to study the effect of fan-shaped mode agitator on electric field uniformity (EF) and simulated the three-dimensional electric field distribution by the finite element (FEM). By comparing models with mode agitation and no mode agitation, eight of the ten models were reduced in coefficient of variation (COV), with a reduction of between 2% and 20% [14]. The COV is used to measure the non-uniformity of the temperature distribution, the smaller the value, the better the uniformity of heating. Geedipalli S.S.R. and Rakesh V. studied the temperature distribution of materials placed in a microwave oven with a turntable, pointed out the unevenness of the temperature distribution and studied the degree of unevenness by using a mode stirrer to increase heating uniformity [15].

Janusz S. et al. performed a microwave-fluidized bed drying experiment on carrot slices. The whole set of drying devices has four microwave feed ports, the microwave power of the two feed ports is fixed, and the other two feed ports vary from 50 W to 650 W. The electric field distribution is more uniform by the geometrical distribution of the microwave feed ports [16]. Zhu H.C., He J.B. et al. proposed a new microwave heating rotating radiation structure. By rotating the microwave feed port at a certain speed, the electric field mode in the cavity is changed to perform mode stirring. In comparison with two heating methods, heating with turntable and direct static materials heating, has shown that the rotating microwave feed method has a better heating uniformity [17]. The heating uniformity is improved by changing the microwave feed port.

Kashyap S.C. and Wyslouzil W. proposed a method of sweeping the microwave frequency through a range to improve the heating uniformity of the microwave oven. The test on the microwave oven proved the feasibility of this method [18]. Bows J.R. et al. suggest that materials can be heated at different frequencies to improve uniformity. Additionally, the feasibility was verified by experiments based on eight discrete frequencies between the combination of 2.4 and 6.2 GHz [19]. The heating uniformity is improved by changing the frequency of the microwave.

This paper proposes a novel frequency sweep method to improve the microwave heating uniformity. Based on injection-locking magnetron, the microwave source is established. The output frequency of the magnetron varies with the frequency of the injected signal. In Section 2, a mathematical model of the frequency-sweep heating process is built and simulated through the combination of the finite element method (COMSOL) and MATLAB programming. In order to validate the simulation results, the experiment is carried out. In Section 3, the analysis of the influence of frequency sweep interval and frequency-sweep bandwidth on the heating uniformity and heating efficiency are discussed.

## 2. Methodology

### 2.1. Multiphysics Simulation

#### 2.1.1. Geometry

The simulation model consists of a WR340 waveguide and a multimode cavity, as shown in Figure 1. Microwaves are fed through the WR340 waveguide with the dimension of 86.4 mm × 43.2 mm. The heated object is a rectangular parallelepiped potato that is placed on the center of the Teflon bracket on the central bottom of the cavity. The model structure is built in a multi-physics software, COMSOL Multiphysics (5.3a, COMSOL Inc., Stockholm, Sweden).

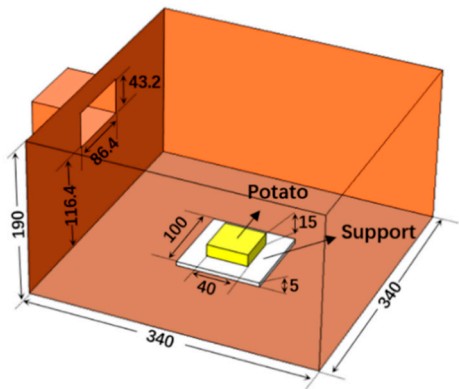

**Figure 1.** Geometry of the 3-D simulation model (unit: mm).

The length of the waveguide is set as half of the wavelength of the waveguide at the simulation frequency, which can be approximated by the WR340 waveguide used in actual experiments [20]. The length of the waveguide is set to 61.22 mm, which can be expressed by

$$Len = \lambda \div \sqrt{1 - \left(\frac{\lambda}{2a}\right)^2}, \ \lambda = \frac{c}{f}, \tag{1}$$

where $\lambda$ is the wavelength of the free space, $a$ is the broadside width of the WR340 standard waveguide, $c$ is the speed of light, and $f$ is the frequency of the electromagnetic wave.

#### 2.1.2. Governing Equations

The simulation of the model is coupled by the electromagnetic field and the heat transfer. In order to calculate the electromagnetic field, Maxwell's equations are used [20].

$$\begin{aligned} \nabla \times \vec{H} &= \vec{J} + \varepsilon \frac{\partial \vec{E}}{\partial t}, \\ \nabla \times \vec{E} &= -\frac{\partial \vec{B}}{\partial t}, \\ \nabla \cdot \vec{B} &= 0, \\ \nabla \cdot \vec{D} &= \rho_e, \end{aligned} \tag{2}$$

where $\vec{H}$ is magnetic field intensity, $\vec{J}$ is ampere density, $\vec{E}$ is the electric field strength, $t$ is time, $\vec{B}$ is magnetic induction intensity, $\vec{D}$ is electric displacement vector, and $\rho_e$ is electric charge density. Wave equation of electric field can thus be derived from Equation (2) and be written as Helmholtz equation [20].

$$\begin{aligned} \nabla \times \mu_r^{-1}\left(\nabla \times \vec{E}\right) - k_0^2\left(\varepsilon_r\varepsilon_0 - \frac{j\sigma}{\omega}\right)\vec{E} &= 0, \\ k_0 &= \omega\sqrt{\varepsilon_0\mu_0}, \end{aligned} \tag{3}$$

where $\mu_r$ is the relative permeability, $k_0$ is the wave number in free space, $\varepsilon_r$ is the relative permittivity, $\omega$ is the angular frequency, $\varepsilon_0$ is the permittivity of vacuum, $\sigma$ is the electrical conductivity, and $\mu_0$ is the permeability of vacuum.

Then the electromagnetic power loss $Q_e$ can be gained from the computed electric field by [21]:

$$Q_e = \frac{1}{2}\omega\varepsilon_0\varepsilon'' \left|\overrightarrow{E}\right|_2, \tag{4}$$

where $\varepsilon''$ is the imaginary part of relative permittivity of processing material.

To compute the temperature distribution, governing equation for heat transfer is given as [22]:

$$\rho C_p \frac{\partial T}{\partial t} - k\nabla^2 T = Q = Q_e, \tag{5}$$

where $\rho$ is the material density, $C_p$ is the material heat capacity, $T$ is the temperature, $Q$ is the heat source, and $k$ is the thermal conductivity.

### 2.1.3. Boundary Conditions

In the simulation, the surface of the entire heating cavity is defined as an ideal electrical conductor except the microwave feed port. The boundary equation can be expressed as [20]:

$$\overrightarrow{n} \times \overrightarrow{E} = 0, \tag{6}$$

where $\overrightarrow{n}$ is the unit normal vector of the corresponding surface. The microwave feed port is defined as a rectangular wave port, and the propagation mode of the microwave in the waveguide is defined as $TE_{10}$. Electromagnetic waves enter the cavity and transform into multiple modes.

We apply the convection heat transfer to the surface of the potato. Here, we assume that the heat of potato convects with the air in the cavity [22].

$$-k \cdot \frac{\partial T}{\partial n} = h \cdot (T - T_{air}), \tag{7}$$

where $T_{air}$ is the temperature of the air and $h$ is the heat transfer coefficient with the value of $10 \, W/(m^2 \cdot K)$. The thermal boundary between potato block and the support disk is set as insulation boundary conditions.

### 2.1.4. Input Parameters

In this model, the frequency of the electromagnetic wave varies from 2.43 GHz to 2.45 GHz, and the electromagnetic wave power input to the rectangular waveguide is 50 W. The entire cavity is filled with air. The initial temperature of the potato block is 293.15 K. It is assumed that the temperature of air in the cavity is also 293.15 K. Since the temperature of the potato block and the frequency of the microwaves are all in a small range, the influence of temperature and frequency on the dielectric constant of the potato block is ignored. Related input parameters of simulation are shown in Table 1 [23].

**Table 1.** Related input parameters.

| Property | Domain | Value | Unit |
|---|---|---|---|
| Relative permittivity | Air | 1 | - |
| | potato | 57–17j | - |
| | PTFE | 2.3 | - |
| Relative permeability | Air | 1 | - |
| | potato | 1 | - |
| | PTFE | 1 | - |
| | Aluminum | 1 | - |
| Conductivity | Air | 0 | S/m |
| | potato | 0 | S/m |
| | PTFE | 0 | S/m |
| | Aluminum | $3.774 \times 10^7$ | S/m |
| Heat conductivity coefficient | potato | 0.648 | W/m·K |
| Density | potato | 1050 | kg/m$^3$ |
| Heat capacity at constant pressure | potato | 3640 | J/kg·K |
| Dielectric loss tangent | potato | 0.298 | - |

### 2.1.5. Frequency Sweep Methods

In order to change the microwave frequency in the model, MATLAB programming (2015a, MathWorks Inc., Natick, MA, USA, 2015) and finite element method (COMSOL) are combined. In each time step, the microwave frequency is updated and the electromagnetic field is calculated in the frequency domain. Then, the temperature is calculated in the time domain in this time step. In the next time step, the results of the previous step are set as the initial value, thus completing the frequency-sweep simulation calculation.

### 2.2. Magnetron Frequency Sweep Output

#### 2.2.1. Equivalent Circuit Model

In order to build up the frequency sweep microwave power source, the injection frequency-locking technique is used. The equivalent circuit model of magnetron locked by an external signal is shown in Figure 2. According to the Kirchhoff voltage law, we can get:

$$-(g + jb)\widetilde{V}_{RF} = \frac{\widetilde{V}_{RF}}{R} + \frac{\widetilde{V}_{RF}}{j\omega L} + j\omega_m C\widetilde{V}_{RF} + C\omega_0 \frac{G + jB + \rho_m e^{-j\theta}}{Q_{ext}}\widetilde{V}_{RF} \tag{8}$$

where $g, b, G, B$ are the equivalent conductance and equivalent susceptance of the magnetron source and load respectively, $R, L, C$ are equivalent resistance, equivalent inductance, equivalent capacitance respectively, $\omega_m$ is the oscillation frequency of magnetron, $\omega_0$ is the resonant frequency of the resonator. $\widetilde{V}_{RF}$ is the RF voltage of magnetron, $\rho_m$ is the ratio of the injection signal voltage to the magnetron RF voltage with the expression of $\rho_m = \sqrt{P_i/P_{RF}}$, $P_i$ is the power of injection signal, $P_{RF}$ is the power of magnetron RF output, and $\theta$ is the relative phase difference between the phase of the external signal and the phase of the RF output [24–26].

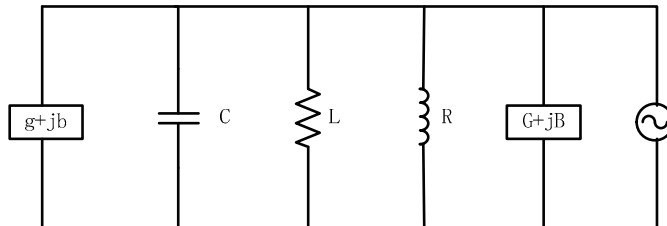

**Figure 2.** The equivalent circuit model of magnetron is being locked by an external signal.

By decoupling Equation (8) and separating the real part and imaginary part, we can obtain the RF output of the magnetron:

$$V_{RF} = \frac{V_{dc}}{1 + C\omega_1 R/Q_L} \frac{1}{\beta} + e^{-\frac{1}{\beta}\gamma t}, \quad Q_L = \frac{1}{\frac{1}{Q_0} + \frac{G}{Q_{ext}}}, \quad \beta = 1 + \rho_m \omega_1 \cos\theta/(\gamma Q_{ext}), \tag{9}$$

and

$$\frac{d\theta}{dt} + 1 - \omega' = \frac{\rho_m}{2Q_{ext}} \sin\theta, \tag{10}$$

where $V_{dc}$ is anode voltage of magnetron, $\omega_1$ is the local oscillator frequency of magnetron resonant cavity, $Q_0$ is inherent quality factor, and $Q_{ext}$ is external quality factor of the resonant circuit and $\omega'$ is the normalized center frequency of the external signal.

While the magnetron is locked by an external signal [27], $\theta$ will be a constant, which means $\frac{d\theta}{dt} = 0$. Therefore, Equation (10) becomes

$$1 - \omega' = \rho_m \sin\theta/(2Q_{ext}), \tag{11}$$

When the external signal frequency is swept in a limited range and $\omega'$ is close to center frequency of free running magnetron, the magnetron can be completely locked by the external signal. In this case, the output signal of magnetron is a sweep frequency signal. If $2Q_{ext}|1 - \omega'|$ is slightly larger than $\rho_m$, the output of magnetron will contain plenty of side-band signals at the integer multiple of the difference between center frequency of the free running magnetron and $\omega'$, $\Delta\omega$ [28]. Therefore, when the external sweep frequency signal moves to this section, the magnetron will be quasi-locked. In this case, the output frequency of magnetron can sweep to this interval. However, there would be some side-band signals at the integer multiple of $\Delta\omega$.

2.2.2. Experimental Setup

In order to verify the accuracy of the simulation results, a microwave sweep heating system was established. The heating system consists of two parts: magnetron injection frequency locking and frequency sweep heating, as shown in Figure 3. The solid-state scan source can provide an injection signal that is injected into the magnetron through two circulators. The input and output power are monitored in real time by a directional coupler and power meter. The locked magnetron signal is sent to the reaction chamber through the circulator to heat the test object.

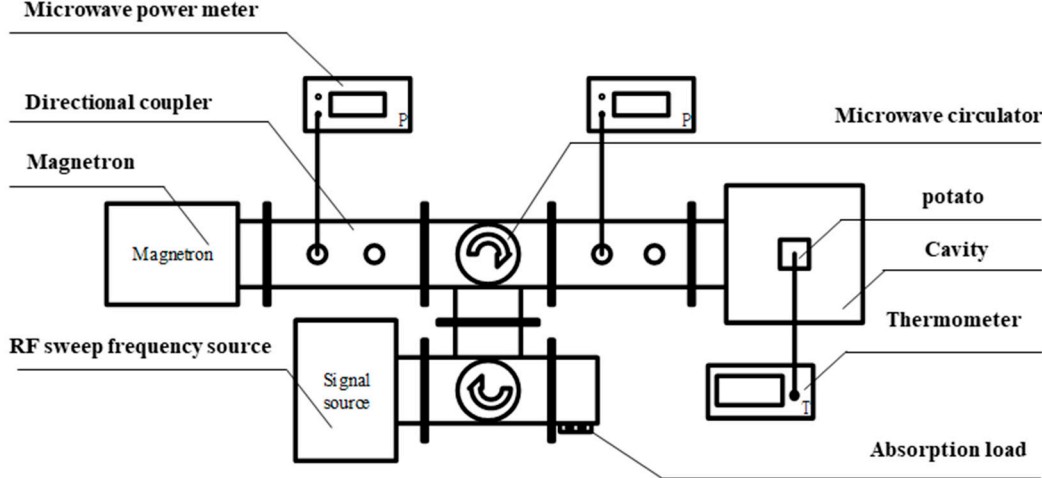

**Figure 3.** Magnetron injection frequency lock heating system block diagram.

The temperature of the potato is tracked by a fiber optic thermometer. A fiber is placed at the center of the upper surface of the potato. A thermal imager is used to measure the temperature distribution of the potato surface. The correctness of the simulation model is verified by comparing with the simulated upper surface temperature distribution and the center point temperature.

### 2.2.3. Magnetron Injection Lock Time

By analyzing the equivalent circuit model of the magnetron, it can be known that there is an equivalent distributed inductance and capacitance, so the magnetron frequency may not keep up with the fast change of the frequency of the external signal. The frequency-time relationship between the magnetron and the externally injected signal was measured by a real-time spectrum analyzer (RTSA). The measurement results are shown in Figure 4. One Panasonic CW magnetron 2M244-M1 is applied in the system to provide high power microwave. A DDY10-5K/0V8-S220/F02 DC power supply from Sichuan Injet Electric Co. Ltd. (Deyang, China) is used to provide high voltage direct current to the magnetron. The external injection signal is provided by a solid-state source. The first row gives a plot of the frequency of the external injection signal over time, and the second row gives a plot of the frequency of the magnetron output signal over time, which is obtained using the frequency time analysis function of the real-time spectrum analyzer. It can be seen that the magnetron output and the external signal are consistent when the sweep period is from 1580 us to 90 us. The injection lock time is negligible here.

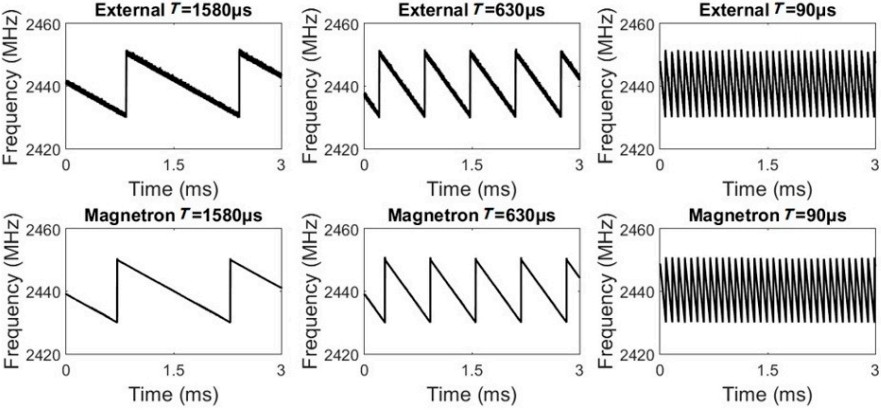

**Figure 4.** The Freq. vs. Time comparison figure of signal source and magnetron with different sweep rate.

### 2.2.4. Magnetron Sweep Frequency Output

According to the above theory and experiment, the sweep frequency output of the magnetron is feasible. The output signal of the above system is analyzed, and the DPX spectrum (Digital Phosphor Spectrum) of the output signal shown in Figure 5. The sweep period is 1580 us and the sweep interval is 5 kHz. The power of external signal is fixed at 52.4 W, the power of magnetron is fixed at 397 W, and sweep frequency ranges from 2430 MHz to 2450 MHz. The DPX spectrum shows that the output signal of magnetron sweeps in the interval smoothly. Hence, the magnetron can be served as a sweep frequency microwave source. The final sweep output total power is about 397 W.

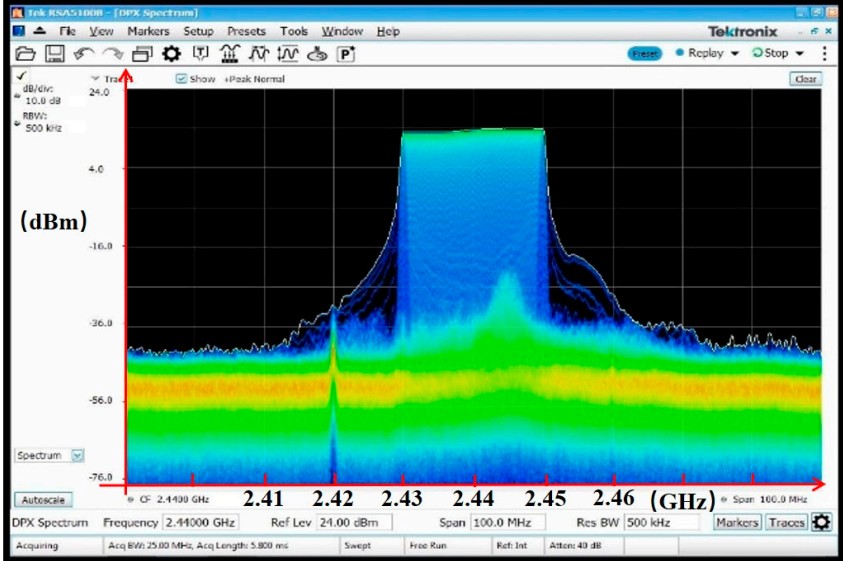

**Figure 5.** The DPX spectrum of magnetron.

## 3. Results and Discussions

### 3.1. Model Validation

In order to verify the feasibility of the simulation model, the above-mentioned injection frequency-locked magnetron was used for the sweep frequency and fixed frequency heating experiments. The total microwave power of both is 397 W. The microwave frequency of fixed frequency heating is 2450 MHz. The sweep cycle of the sweep frequency heating is 1580 us, the frequency interval is 5 kHz, and the sweep frequency ranges from 2430 MHz to 2450 MHz. A potato with a size of 50 cm × 50 cm × 15 cm is used for the 20 s heating experiment.

The upper surface temperature distributions of the simulation and experiment after 20 s heating have been showed in Figure 6. It is not difficult to see that the results of the experiment are almost identical to the results of the simulation. The location of the hot spot distribution and the shape of the hot spot in both experimental results are consistent with the simulation results. The area of the hot spot in the experimental results is slightly larger than the simulation result. This may be due to simplified physical models and conditions during the simulation. The physical process of heating potato includes not only heat transfer and convection, but also changes in moisture in the potato and the flow of air in the chamber. These simplifications may cause errors between simulation and experiment.

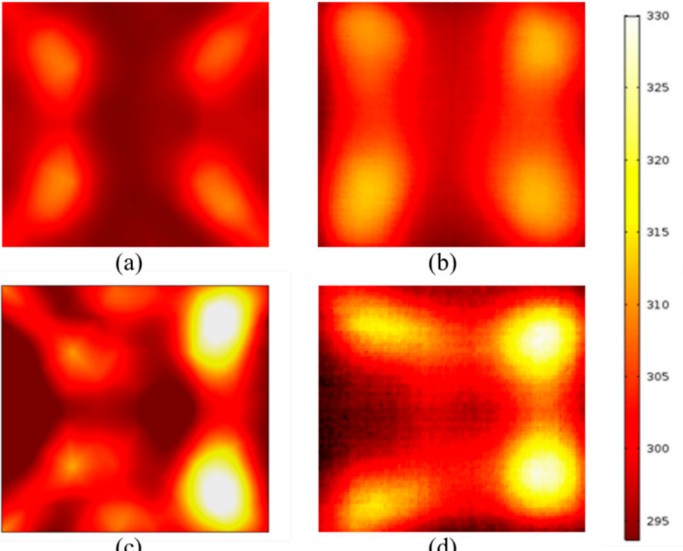

**Figure 6.** Experiment and simulation results of potato surface temperature distribution (unit: K):
(**a**) Simulated heating results of the fixed frequency heating; (**b**) experimental heating results of the fixed
frequency heating; (**c**) simulated heating results of the sweep frequency heating; and (**d**) experimental
heating results of the sweep frequency heating.

The curves of the temperature at the center point on the upper surface of the potato as a function
of time during the sweep frequency heating simulation and experiment are shown in Figure 7.
The experimental results almost agree well with the simulation results. They have the same trend and
have almost the same temperature rise at the same time. However, the two curves do not completely
coincide. The reason may be that during the simulation, the dielectric constant is not calculated as a
function of temperature and frequency due to the simplified model. While this will bring a little error,
the simplified model reduces the calculation and increases the calculation speed.

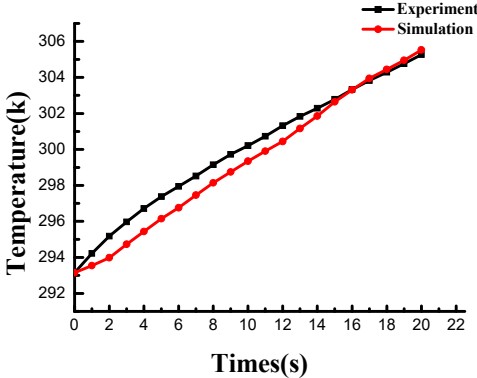

**Figure 7.** Simulation and experimental study on the temperature change of the upper surface
center point.

## 3.2. Heating Uniformity Analysis

The coefficient of temperature variation (COV) is used to measure the non-uniformity of the
temperature distribution [17]. The calculation method is the standard deviation of the temperature rise

at each position of the sample divided by the average value of the temperature rise at each position. The COV could be expressed as:

$$\text{COV} = \sqrt{\frac{\sum_n (T_i - T_a)^2}{n}} / (T_a - T_0) \tag{12}$$

where $T_i$ is the point temperature of the selected region, $T_a$ is the average temperature of the selected region, $n$ is the total number of the point of it, and $T_0$ is the initial average temperature.

In order to analyze the heating uniformity, the microwave power is set as 397 W, the sweep frequency bandwidth is set as 20 MHz, and the frequency interval is set as 5 kHz. There are 4001 frequency points for heating. Since the dielectric constant of the potato block is ignored as a function of temperature and frequency, the effect of the sweep cycle on the simulation results is negligible. The frequency sweep cycle will not be discussed later. The heating time is 20 s and the initial temperature of the potato block is 293.15 K. Figure 8 shows the temperature distribution of the different sections of the fixed frequency and sweep frequency heating. The cut surface of the potato block is from top to bottom. By comparing the temperature distribution of fixed-frequency heating and sweeping frequency heating, we can find that the temperature distribution of the sweep heating is more uniform, which can be seen in Table 2. Sweeping frequency heating not only increases heating efficiency, but also reduces COV, which means that sweep frequency heating increases the uniformity of heating.

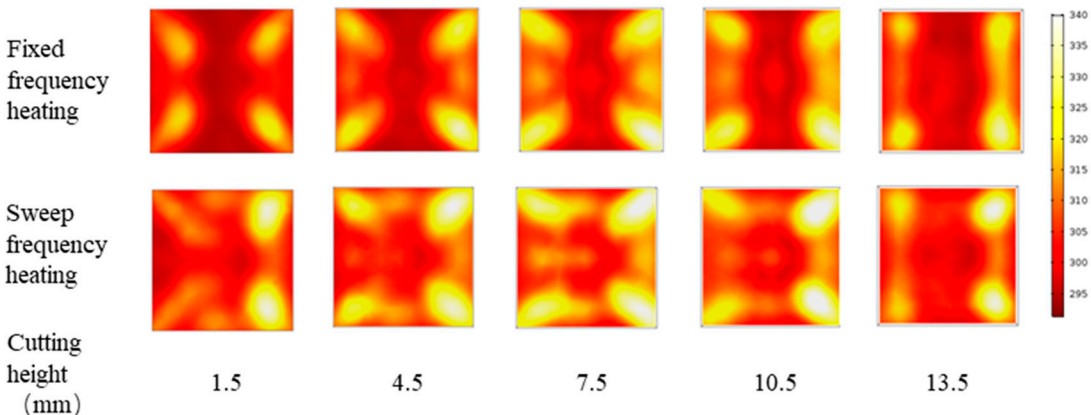

**Figure 8.** Simulation results of different profile temperature distributions between frequency sweep and fixed frequency heating (unit: K).

**Table 2.** Sweep frequency and fixed frequency heating simulation results.

| Heating Method | Average Body Temperature | COV |
|---|---|---|
| Fixed frequency heating | 302.33 K | 0.699 |
| Sweep frequency heating | 311.24 K | 0.535 |

*3.3. Electric Field Analysis*

The appearance of hot spots during microwave heating is an important factor affecting heating uniformity. The cause of the hot spot is that a certain position of the heated object is under a strong electric field for a long time. Therefore, monitoring changes in the electric field can help to analyze the uniformity of heating. Figure 9 below shows the electric field distribution at different frequencies. It is not difficult to see that the electric field changes during the sweep frequency heating process, the peak of the electric field strength gradually shifts from the left side of the potato to the right side of the potato. The possibility of hot spots is greatly reduced, and the uniformity of heating is improved.

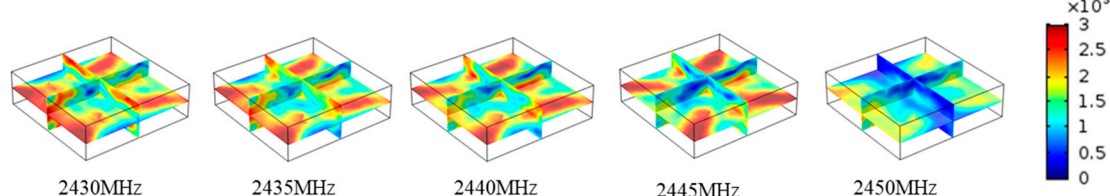

**Figure 9.** The electric field distribution at different frequencies.

### 3.4. Simulation Model Applicability Verification

Material with different dielectric coefficients was used for simulation verification under the above conditions. Pears and potatoes have different dielectric loss tangent, so pears are selected for simulation and experimentation. The simulation input parameters of pear are shown in Table 3 [29], the simulation results are shown in Table 4, and the experimental results are compared, as shown in Figure 10.

**Table 3.** Related input parameters of pear.

| Property | Value | Unit |
|---|---|---|
| Relative permittivity | 64–13j | - |
| Heat conductivity coefficient | 0.648 | W/m·K |
| Density | 1050 | kg/m$^3$ |
| Heat capacity at constant pressure | 3640 | J/kg·K |
| Dielectric loss tangent | 0.203 | - |

**Table 4.** Sweep frequency and fixed frequency heating simulation results.

| Heating Method | Average Body Temperature | COV |
|---|---|---|
| Fixed frequency heating | 300.76 K | 0.634 |
| Sweep frequency heating | 314.66 K | 0.473 |

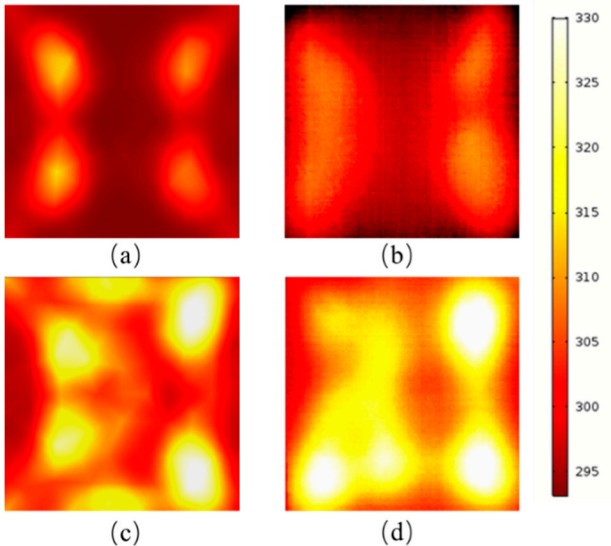

**Figure 10.** Experiment and simulation results of pear surface temperature distribution (unit: K): (**a**) Simulated heating results of the fixed frequency heating; (**b**) experimental heating results of the fixed frequency heating; (**c**) simulated heating results of the sweep frequency heating; and (**d**) experimental heating results of the sweep frequency heating.

From the simulation and experimental results of the heating pear, it is easy to see that the sweep frequency heating can improve the uniformity. The upper surface temperature distribution of the

simulation and the experiment is roughly consistent. This not only proves the reliability of the simulation model again, but also proves that the simulation model can be applied to other materials with different dielectric constants.

### 3.5. Parameter Analysis Affecting Heating Uniformity

### 3.5.1. The Effect of Sweep Bandwidth

Heating simulation at different scanning bandwidths with constant center frequency 2440 MHz is performed. The potatoes were used as a heating simulation model. The models with a sweep frequency of 20 MHz, 40 MHz, 60 MHz, and 80 MHz bandwidths are simulated, and the sweep interval is set as 5 kHz. The larger the bandwidth, the more frequency points are calculated. When the bandwidth is 80 MHz, there are 16,001 frequencies for heating. The heating time is 20 s, and the initial temperature is 293.15 K. The heating temperature distribution are given in Figure 11. The average temperature and COV comparison are shown in Table 5. By comparing the temperature distribution of heating at different bandwidths, we can find that the potato block has a very optimistic temperature distribution with a wider sweep frequency bandwidth. Therefore, it can be concluded that, within a certain range, the wider the sweep bandwidth, the higher the uniformity of heating. This may be because that a wider sweep bandwidth produces more electric field distribution and the electric field changes more drastically during heating. However, the heating uniformity and average temperature may be no longer improved with a wider sweep bandwidth which can be seen the results with the bandwidth of 60 MHz and 80 MHz, since the variation of electric field distribution may be saturated in the sample. There are two reasons account for the phenomenon. First, the shape of the object to be heated causes a problem of focusing of the electromagnetic field. For example, a spherical object tends to focus the energy of the electromagnetic field at a central position, and heating of the spherical material tends to cause hot spots in the center. Second, in the frequency of heating, some part of the new frequency does not necessarily contribute to improving uniformity.

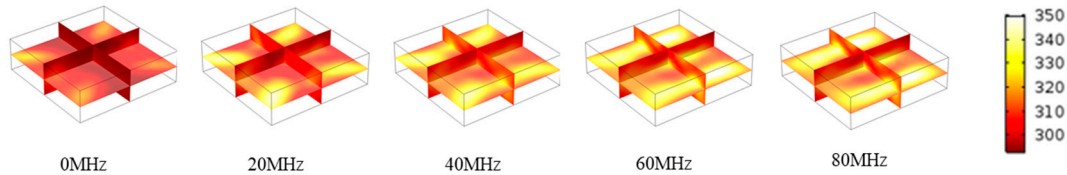

**Figure 11.** The heating temperature distribution under different sweep frequency bandwidths.

**Table 5.** Simulated heating results for different sweep bandwidths.

| Sweep Frequency Bandwidth | Number of Frequency Points | Average Body Temperature | COV |
|---|---|---|---|
| Fixed frequency | 1 | 302.33 K | 0.699 |
| 20 MHz | 4001 | 311.24 K | 0.535 |
| 40 MHz | 8001 | 314.35 K | 0.413 |
| 60 MHz | 12,001 | 317.70 K | 0.373 |
| 80 MHz | 16,001 | 317.41 K | 0.369 |

### 3.5.2. The Effect of Sweep Interval on Uniformity

Further study on the effect of different sweep frequency intervals on heating uniformity under a fixed sweep frequency bandwidth is performed. The potatoes were used as a heating simulation model. The sweep range is from 2430 MHz to 2450 MHz, the heating time is 20 s, and the initial temperature is 293.15 K. The heating results with different sweep intervals are compared in Table 6. It can be seen that during the gradual decrease of the sweep interval, the average body temperature and COV gradually become stable. This stable value occurs when the frequency interval is 200 kHz. The effect of the sweep

interval on heating efficiency and uniformity is less than the effect of bandwidth on it. Frequency interval is an important parameter affecting heating uniformity, but it does not completely determine heating efficiency and heating uniformity. The COV is only 0.469 when the frequency interval is 5 MHz. It is shown that the combination of these five frequency points is beneficial to increase the heating uniformity. Analyzing the sweep interval helps us find the most suitable combination of heating frequency points.

**Table 6.** Simulated heating results for different sweep intervals.

| Sweep Interval | Number of Frequency Points | Average Body Temperature | COV |
|---|---|---|---|
| 5 kHz | 4001 | 311.24 K | 0.535 |
| 20 kHz | 1001 | 311.28 K | 0.531 |
| 200 kHz | 101 | 311.29 K | 0.528 |
| 2 MHz | 11 | 310.51 K | 0.522 |
| 5 MHz | 5 | 310.56 K | 0.469 |
| 20 MHz | 2 | 309.24 K | 0.576 |

### 3.5.3. Effect of Dielectric Coefficient on Sweep Frequency Heating

The effects of sweep heating on different materials were studied by simulating materials with different dielectric constants. Two sets of simulations were performed. The first set of simulations only changed the real part of the dielectric constant. The second set of simulations only changed the imaginary part of the dielectric constant. The other input parameters were consistent with the potato heating model parameters in Section 3.1. The simulation results of the imaginary part of the dielectric constant were shown in Table 7, and the imaginary part of the dielectric constant was 17. The simulation results of the real part of the dielectric constant were shown in Table 8, and the real part of the dielectric constant was 57.

**Table 7.** Simulated heating results for different real part of Dielectric coefficient.

| Heating Method | Real Part of Dielectric Constant | Dielectric Loss Tangent | Average Body Temperature | COV |
|---|---|---|---|---|
| Fixed frequency heating | 20 | 0.850 | 310.53 K | 0.547 |
|  | 40 | 0.425 | 299.61 K | 0.538 |
|  | 60 | 0.283 | 301.64 K | 0.616 |
|  | 80 | 0.213 | 312.41 K | 0.520 |
| Sweep frequency heating | 20 | 0.850 | 315.75 K | 0.397 |
|  | 40 | 0.425 | 316.06 K | 0.513 |
|  | 60 | 0.283 | 316.36 K | 0.518 |
|  | 80 | 0.213 | 305.74 K | 0.600 |

**Table 8.** Simulated heating results for different Imaginary part of dielectric constant.

| Heating Method | Imaginary Part of Dielectric Constant | Dielectric Loss Tangent | Average Body Temperature | COV |
|---|---|---|---|---|
| Fixed frequency heating | 5 | 0.087 | 301.08 K | 0.650 |
| | 10 | 0.175 | 303.20 K | 0.664 |
| | 15 | 0.263 | 304.01 K | 0.688 |
| | 20 | 0.351 | 304.23 K | 0.732 |
| Sweep frequency heating | 5 | 0.087 | 311.60 K | 0.585 |
| | 10 | 0.175 | 315.00 K | 0.607 |
| | 15 | 0.263 | 315.27 K | 0.563 |
| | 20 | 0.351 | 315.00 K | 0.542 |

It can be seen from the simulation results that the sweep frequency heating can effectively improve the uniformity of heating. However, in the eight sets of simulations, a set of sweep heating resulted in poorer uniformity than fixed-frequency heating, as shown in Table 7. It may be because that the heated load constant matches better with the electromagnetic wave transmission line at 2.45 GHz. Therefore, better energy absorption is obtained. It may be concluded that sweep frequency heating can increase the uniformity of heating in most cases, and the best heating performance needs to be discussed in a practical situation.

## 4. Conclusions

In this paper, a method for microwave sweep frequency output of magnetron is proposed is proposed. The sweep frequency output of magnetron is realized by using injection-locking frequency theory. The influence of the magnetron power injection ratio on the scan bandwidth and the influence of the sweep speed on the lock time are analyzed. The injection frequency-locked system is built to realize the sweep frequency output of the magnetron. A general model for calculating microwave sweep frequency heating is established. Multiphysics calculation based on the finite element method is carried out with COMSOL Multiphysics software. A sophisticated experimental system is developed to validate the multiphysics model. The swept heating test of the magnetron injection frequency is completed by using the potato block. The results show that the frequency sweep heating effectively improves the uniformity and efficiency. The COV of the potato heated by sweep frequency was reduced from 0.699 to 0.535. Additionally, the experimental results are consistent with the simulation results. This paper also analyzes the important parameters that affect the uniformity of sweep frequency heating, which are the sweep bandwidth and the sweep interval. It can be concluded that within a certain range, the wider the sweep bandwidth, the higher the uniformity of heating. However, as the bandwidth increases, the uniformity of heating may be saturated. When the sweep range is determined, the smaller sweep interval has better heating uniformity and efficiency. This conclusion is not absolute. Some sweep intervals may lead to better combinations of frequency points, and heating at these frequencies will also show good uniformity.

**Author Contributions:** H.Z. conceived and designed the experiments; F.Y. developed the model, performed the experiments, analyzed the data, and wrote the initial draft of the manuscript; W.W., B.Y., T.H., Y.Y. and L.W. reviewed and contributed to the final manuscript; K.H. contributed the location and equipment.

**Funding:** This work was supported by the NSFC—China (Grant No. 61731013), Science Foundation of the Guizhou Province (Grant No. Qiankehezhicheng [2018] 2004), and the Special Fund of Strategic Cooperation between Sichuan University and Luzhou City (Grant Nos. 2017CDLZ-G12).

**Conflicts of Interest:** The authors declare no conflict of interest.

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
