# Peer review of "Sweep Frequency Heating based on Injection Locked Magnetron"

_processes, doi:10.3390/pr7060341_

Round 1
Reviewer 1 Report
It is an interesting idea to use the injection locking phenomenon of the magnetron to make the microwave heating uniform. However, if it is 500 W, there is already a semiconductor oscillator. Everyone doesn’t use magnetron using low power application of microwave heating.
- It takes time until injection locking, but there is no description. Therefore, the argument of 3.4.2 is useless.
- Although the frequency sweeping method is described for electromagnetic wave and heat transfer calculation using COMSOL and MATLAB, there is no explanation of the number of steps. Under 50 kHz frequency sweep, how many calculations during 1 sec was carried out? In other words, how many frequencies were employed for heat transfer? Authors showed only five electric fields. Authors should show how many frequencies are enough for obtaining the heat transfer results before discussion of COV.
- Figure 4 should be rewritten and low quality. What do the two drawings on the left side of Figure 4 represent? May they depend on the synchronization of DPX spectrum analyzer? It takes some time to lock the signal of magnetron and it also depends on external power. There is no meaning unless the measurement should be accumulated. Also, the vertical axis and the horizontal axis of the right side of Figure 4 cannot be seen.
Author Response
Dear Reviewer:
We appreciate your hard work on our manuscript. Thanks very much for your excellent suggestions. According to your suggestions and questions, we have made the following changes in the latest version of the manuscript.
Best regards!
Huacheng Zhu & All of other co-authors.

Reviewer 2 Report
This paper shows that heating uniformity in a microwave oven can be improved by frequency-sweeping the microwave source – an interesting and potentially very useful alternative to mode-stirring with a mechanical paddle.
The paper is logically structured, and the methods and results are described clearly. It’s good to see detailed results of simulations backed up with carefully conducted experiments.
If possible, please explain why the effect saturates with increasing bandwidth, and shows little further improvement beyond 80MHz. Would this saturation bandwidth be different if the cavity dimensions, or the size and shape of the sample, changed?
Also, would a higher sweeping bandwidth be practically achievable with your experimental set-up?
Line 54: please define COV here (rather than on line 224)
Lines 88-91 you explain how you calculated ‘Len’ but what is the actual value?
Line 90: ‘a’ should be ‘broadside width’ rather than ‘broadside length’
Eq. 7 is confusing – is the ‘a’ the same as in eq. 1? Please check.
Table 1, penultimate line, kg/m3 should have superscript 3
Author Response

(The authors gave the same response as above.)

Reviewer 3 Report
Dear Authors,
I read your paper, and there are my comments:
The English fluency of your work needs several revisions.
The abstract must be improved.
The introduction is too poor.
Your result needs more discussions.
Please write your paper based on the template of this journal. For example, the font and size of the conclusion are wrong.
In figure 6, the simulation results maybe cannot follow the experimental results for after 20 seconds. Therefore, could you increase the simulation time and sample rate?
In my opinion, your paper will be suitable for publishing in this journal after revisions.
Sincerely yours,
Author Response

(The authors gave the same response as above.)

Round 2
Reviewer 1 Report
It is important to change frequecy to obtain temperature uniformity although ISM band is allowed. Frequency, in other word, is propotinal to 1/wavelength. In ISM band, we can change wavelength just 5%, means 100MHz. So, result of COV is not drastrically improved due to small change of the wavelenght. Usually, stair fan is equipped in mutlimode applicator. The authors employed the potato as a heating sample to obtain the small COV due to small range of output frequecy.
(1)Please try to another materials which have different permittivety and tangent delta.
(2)The value of the vertical axis in Fig. 7 is wrong, and the caption of Fig. 7 was Inappropriate. Is the tempereature differece between max. and min. on the surface?
Author Response
Dear Reviewer:
We appreciate your hard work on our manuscript. Thanks very much for your suggestions. According to your suggestions and questions, we have performed new simulations and experiments. And adding two sections in the article to illustrate this problem.The specific changes are shown in the attachment.
Best regards!
Huacheng Zhu & All of other co-authors
